# Feasibility of using an energy balance to measure evapotranspiration in green stormwater infrastructure

**Devin Thomas Krasowski** (ID)*, **Bridget Wadzuk, Brian Jacko**

Department of Civil and Environmental Engineering, Villanova Center for Resilient Water Systems, Villanova University, Villanova, Pennsylvania, United States of America

* devink1331@gmail.com

## Abstract

Effective green stormwater infrastructure (GSI) design requires comprehensive quantification of the volume of water that can be treated or removed over a given time period. It is recognized that evapotranspiration (ET) can be a substantial pathway for stormwater volume reduction in bioretention systems. However, measuring ET is often difficult and expensive, such as with lysimeters or a mass balance approach. This research focused on a new technique for quantifying ET in bioretention systems by exploring an approach using thermal imaging to calculate ET by measuring the flux of energy at the canopy surface. This thermal imaging approach was compared to ET measurements given by a traditional mass balance approach. The experimental setup had three benchtop scale vegetated lysimeters planted with Switchgrass. Time lapse thermal images of the Switchgrass plants were taken at 10 second intervals and paired with meteorological data. The data were used in an energy balance to estimate the mass of water lost from the lysimeter plant/soil system. That mass was compared to the change in weight measured by weighing the lysimeter before and after the data collection period. For comparison, reference ET was also calculated for the vegetated systems using three common reference ET equations. The uncalibrated energy balance equation developed here estimated an averaged ET over 12 data collection days within 1 mm of the mass balance measured ET. These findings demonstrate the feasibility of using a thermal image energy balance technique to estimate ET.

## Introduction

### Background

Understanding evapotranspiration (ET) is important for the design, construction, operation, and maintenance of green stormwater infrastructure (GSI) [1–3]. In a larger context, ET is important for a variety of land use applications. For instance, there is a connection between ET rates and drought conditions [4]. Across the United States, daily average ET rates grouped by county can vary from nearly zero to approximately 2.74 mm/day [5]. It is necessary to contextualize the anticipated ET along with the land use. One land use application that has long

**Data Availability Statement:** All processed data files and original MATLAB code are available from the https://github.com/devink1331/EnergyBalance database.

**Funding:** This work was funded by a grant applied for by and awarded to Dr. Bridget Wadzuk (BW). This internal grant from the Villanova University Department of Civil and Environmental Engineering (https://www1.villanova.edu/university/engineering/academic-programs/departments/civil-environmental.html) was awarded to fund tuition and provide a stipend for one Master's student (DTK) to carry out the research and to provide compensation to undergraduate research assistants (BJ).

**Competing interests:** The authors have declared that no competing interests exist.

understood and investigated methods for quantifying ET is agriculture, where the cost of irrigation, the need to adequately manage water resources, and water shortages make it necessary for farmers to use water efficiently and estimate crop water demand by quantifying ET. In both agriculture and GSI, ET is a substantial portion of the water budget. An alfalfa field and a rain garden can both have ET rates approaching 10 mm/day [1–3, 6, 7]. Thus, the stormwater community can benefit from techniques used in agriculture to better understand and incorporate ET into design.

Urbanization changes the hydrologic cycle so designing infrastructure that mimics the natural hydrologic cycle helps mitigate the resulting adverse effects [8]. As the benefits of GSI become more realized so do the benefits of having them filled with plants to promote more ET [8]. It has been shown via lysimeter studies that ET can account for upwards of 50% of the water balance in GSI [9]. Finding easy and effective methods to estimate ET can encourage efficiency and help inform parameters like plant type, overall GSI size, maintenance requirements, and volume reduction capacity. Evapotranspiration quantification can also lend the industry, specifically regulatory agencies, confidence in allowing for more accurate volume reduction credits for GSI systems [8]. Lastly, accurate ET predictions can aid in developing real time control logic to help make informed reuse and outflow decisions.

Physically and empirically based reference ET equations that have been developed and are widely used to quantify ET in agriculture [9] have also been applied in stormwater [2, 6, 10]. More recently, those equations have been used along with satellite based thermal imaging to quantify ET at large ($\approx$ 100 m) spatial scales and at multi-day time steps.

Green stormwater infrastructure are not always well watered, are generally installed in urban areas, are not uniform, and are usually smaller than 100 m$^2$. Therefore, water balance based methods that require well-watered and homogeneous conditions may not work well in GSI applications, at least not without modification [2, 10]. Energy balance methods that using thermal imaging have not been investigated to calculate ET for GSI. However, thermal imaging is becoming more of a common tool in other sectors for approximating ET, particularly when it is paired with an energy balance [11–13]. As water undergoes the phase transition from liquid near the leaf surface to water vapor in the atmosphere, the energy required to break the hydrogen bonds must come from the leaf surface [14]. If all sources of incoming and outgoing energy are measured or estimated, then the latent energy facilitating the phase change of water can be solved for. Thermal imagery from airplanes, drones, and ground based thermal cameras have been used to calculate ET, plant water stress, and leaf temperature with scales ranging from meters squared to millimeters squared [12, 15]. This leads to the research question, *Is it practical to evaluate plant performance in GSI via thermal imaging and an energy balance*? In this paper, the feasibility of using thermal imaging paired with an energy balance to quantify ET in a GSI application is investigated. The research site construction, data collection, and analytical methods will be discussed, and the results and possible implications for the research community and industry will be synthesized.

## Methods

This study was carried out during the summer of 2019 to evaluate the feasibility of estimating ET from GSI with thermal imaging and an energy balance for three vegetated lysimeter buckets (**Fig 1**, photo taken on 7/19/2020 prior to full data collection days). A mass balance approach using the vegetated lysimeters was also used to measure ET. Reference ET estimated by three different reference ET equations was compared to the mass balance ET results to investigate whether or not the collected meteorological data, when used with already accepted methods, could produce reasonable results. Then, the energy balance calculated ET was

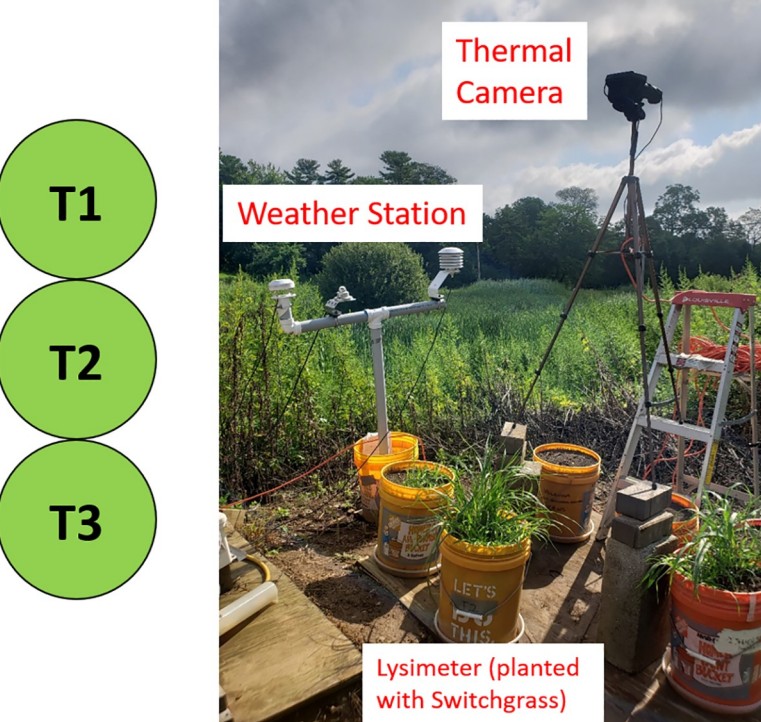

**Fig 1. Thermal imaging study site.** (left) Labeling of the three lysimeter buckets. (right) Example of thermal imaging site set up including the thermal camera, weather station, and lysimeter bucket.

compared to the mass balance measured ET to determine the feasibility of the energy balance approach.

## Site setup

Evapotranspiration for this study was measured, as in Eq 1, via a mass balance from three discrete weighing lysimeter buckets [16]:

$$ET = P - I + \Delta W \tag{1}$$

where, for any time step, $\Delta W$ is the change in weight ($W_0 - W_f$), $W_o$ is the weight at the beginning of the time step and $W_f$ is the weight at the end of the time step), $P$ is the weight of any precipitation that fell, $I$ is the weight of infiltration that exited the column, and $ET$ is the mass balance measured evapotranspiration. Lysimeter buckets were weighed using an A&D GP series scale [17]. Measurements were taken to the 0.012 mm (0.1 g) with an accuracy of ± 0.37 mm (± 3 g).

   Three test lysimeters labeled T1 through T3 (shortened from "Test Setup 1" through "Test Setup 3") contained Switchgrass and were constructed using 5-gallon (19 L) buckets (36.8 cm tall with a 28.6 cm diameter; **Fig 1**). Each of three lysimeters were constructed in the exact same manner. The buckets were filled with 31.5 cm of sandy loam soil collected from Villanova's campus and holes were drilled in the bottom to allow for percolation out of the lysimeter. The Switchgrass plants were grown from seed planted in the spring of 2019. The Switchgrass was allowed to grow naturally in each lysimeter, which resulted in slightly different rates of growth. During each of the full data collection days analyzed, the Switchgrass in T1, T2, and T3 were approximately the same height and shape. Percolated water out of each lysimeter was

captured and weighed along with each lysimeter at the beginning and end of the experimental period. Experiment periods were typically about 5 hours during the middle of the day from July to October 2019.

Instrumentation for the site included a FLIR E95 thermal camera to measure leaf temperature (±2˚C) [18], a Decagon Devices DS-2 sonic anemometer to measure wind speed (±0.30 m/s or ± <3%) [19], a Huskeflux NR01 net radiometer (±10% over 12 hours) [20], and a Vaisala HMP50 air temperature and relative humidity probe (±2˚C, ±3% RH) [21].

## Measurement and analysis

Over the study period, there were 24 days of recorded data. However, full data analysis was only performed on 12 of those data collection days, as net short-wave and net long-wave radiation were only collected after August 28, 2019. For each data collection day, one of the three lysimeters was selected to be the test lysimeter. Each lysimeter was the test lysimeter for approximately four of the twelve full data collection days. The test lysimeter was weighed on the A&D Scale [17]. The FLIR E95 thermal camera was positioned above the test lysimeter and set to take thermal images every 10 seconds [18]. The camera was positioned at approximately the same height and angle relative to the test lysimeter for each data collection day, and care was taken to ensure there were no shadows over the lysimeter. After the experimental period of approximately 5 hours, the lysimeter was weighed again. The weights were used in the mass balance to calculate ET (Eq 1). Ten second meteorological data was also recorded by the weather station shown in **Fig 1**.

## Energy balance

An energy balance was calculated for each lysimeter for the 12 data collection days. The method for constructing the energy balance used in this study is a combination of those found in research from agriculture, biophysics, and plant physiology [15, 22, 23]. To the knowledge of the authors, the present research presents the first application and evaluation of the energy balance approach for estimating ET from GSI using thermal imagery. The goal of this exploratory analysis was to assess the general feasibility of using an energy balance approach to evaluate the performance of GSI.

This study was interested in how the quantity of water leaving a canopy system relates to the energy fluxes of the system. An energy balance equation can be formulated (Eq 2) to include terms representing all the incoming and outgoing energy of the canopy system. Every term in the energy balance can be measured or estimated except for the term representing the energy that goes into evaporating water from the system. This energy related to evaporation corresponds to a water mass (or volume or depth) because the amount of energy per quantity of water required to facilitate the phase change of liquid water to water vapor is a known value (i.e., the latent heat of vaporization) [22].

The overall energy balance for a canopy in the steady state can be written as:

$$0 = Q^+_{SW} + Q^+_{TIR} - Q^-_{TIR} - Q^-_E - Q_C - Q_S \tag{2}$$

where each term is defined and explained as done in Hikosaka et al. (2016).

$Q^+_{SW}$ is the energy flux density of canopy absorbed radiation,

$$Q^+_{SW} = a_s F_s R_s \tag{3}$$

where $a_s$ is the shortwave absorptivity. For this application a single absorptivity was assigned to the entire plant and for the whole waveband of short-wave radiation pertinent to the canopy energy balance (~285 to 2500 nm) [23] and was set at about 0.81 for Switchgrass; $F_s$ is the view

factor as described in the formulation of $Q^+_{TIR}$ (Eq 4) and was approximated to be 1; $R_s$ is incoming shortwave radiation measured by the NR-01 sensor on the weather station [24].

$Q^+_{TIR}$ is the energy flux density of thermal infrared radiation that describes long wave thermal energy being radiated to the canopy from the sky, from molecules in the atmosphere, and from terrestrial bodies near the canopy. Only radiation in the thermal infrared waveband (~3000 to 50000 nm) is considered. [22]. However, not all of the long wave radiation being emitted by the environment surrounding the canopy will be intercepted by the canopy, which requires the concept of "view factors" [23]. Only some fraction of the long wave radiation emitted from the atmosphere, $F_a$, and the ground, $F_g$, will actually be intercepted by the canopy. Accurately estimating view factors for each of the three different lysimeters is a complex task beyond the scope of this study. For the purposes of this study, the view factors were either assigned values using simplified methods or explored as calibration parameters. In addition, not all of the long wave radiation intercepted by the canopy will be absorbed, thus the energy flux also depends on an absorptivity coefficient $a_L$ which for most plants is around 0.96 [22]:

$$Q^+_{TIR} = a_L(F_a L_a + F_g L_g) \tag{4}$$

$Q^-_{TIR}$ is the thermal and radiative infrared flux density emitted from the canopy surface:

$$Q^-_{TIR} = \varepsilon \sigma T^4_{air} + g_r C_{p,air}(T_{canopy} - T_{air}) \tag{5}$$

where $\varepsilon$ = 0.96 is the emissivity of the canopy, $\sigma$ is the Stephan-Boltzman constant (5.67 x $10^{-8}$ Wm$^{-2}$K$^{-4}$), $C_{p,air}$ is the molar heat capacity of air (0.10647 Jmol$^{-1}$K$^{-1}$), $T_{canopy}$ and $T_{air}$ are the temperatures of the canopy and air respectively, and the radiative conductance in the boundary layer, $g_r = (4\sigma T^3_{air})/(C_{p,air})$. The $Q^-_{TIR}$ describes the long-wave energy being emitted from the canopy. It is made up of the first term on the right hand side that is the transfer of kinetic energy as heat from between the canopy and atmosphere [15, 22, 23] and the second term on the right hand side is the radiative long-wave energy emitted from the canopy [23].

$Q^-_E$ is the energy flux density of latent heat:

$$Q^-_E = \lambda E \tag{6}$$

where $E$ is the flux density of water vapor from the leaf and $\lambda$ is multiplied by the latent heat of vaporization, which is about 2,260 kJkg$^{-1}$. Since $\lambda$ varies slightly with air temperature it was estimated for each time step using an empirical relationship from the US Army Corps of Engineers water quality model CE-QUAL-R1 [25]. $E$ was solved for to give the amount of water evaporated for each time step.

$Q_C$ is the convective heat loss to the air (also called sensible heat loss) from the canopy, through the canopy boundary layer, and to the bulk atmosphere. It may be positive or negative:

$$Q_C = g_{b,h} C_{P,air}(T_{canopy} - T_{air}) \tag{7}$$

where the boundary layer conductance for heat $g_{b,h} = 1.4(0.135)(u/d)^{.5}$, $u$ is the wind speed, and $d$ is the characteristic leaf dimension ($d = 0.72$ * estimated width of each leaf in the direction of the wind) [15]. A constant leaf width of 1 cm was used.

$Q_S$ is the storage term that accounts for energy stored in the plant either as thermal energy from abrupt changes in environmental conditions or as chemical energy from photosynthesis. The time steps considered in this study are such that the energy storage term is negligible [22].

Eq ($2$) can be rearranged to solve for E (in mol/m$^2$/s) that can be used as a proxy for system ET assuming the soil is mostly covered with vegetation [26]:

$$\lambda E = Q_{SW}^+ + Q_{TIR}^+ - Q_{TIR}^- - Q_C \qquad (8)$$

**Penman-Monteith reference ET calculations.** To provide a common model comparison, the mass balance measured ET (Eq 1) was compared against ET from a reference surface as calculated by three different common reference ET equations (FAO Penman-Monteith and two forms of the ASCE Standardized Reference Evapotranspiration Equation; Eqs 9–11). Then the reference ET equation that was best formulated for the Switchgrass systems under investigation (Eq 11) was used to further validate the collected meteorological data.

The first reference ET equation evaluated was the FAO Penman-Monteith equation [26], which has been used in agriculture and some stormwater comparisons (e.g. [2, 26, 27]). The FAO Penman-Monteith equation (PM) is:

$$ET_0 = \frac{0.408\Delta(R_n - G) + \gamma\left(\frac{C_n}{T_{hr}+273}\right)u_2\left(e^o(T_{hr}) - e_a\right)}{\Delta + \gamma(1 + C_d u_2)} \qquad (9)$$

where $ET_0$ is the reference evapotranspiration (mm hr$^{-1}$), $R_n$ is the net radiation at the grass surface (MJ m$^{-2}$ hr$^{-1}$), G is the soil heat flux density (MJ m$^{-2}$ hr$^{-1}$), $T_{hr}$ is the mean hourly air temperature (°C), $\Delta$ is the saturation slope vapor pressure curve at $T_{hr}$ (kPa °C$^{-1}$), $\gamma$ is the psychrometric constant (kPa °C$^{-1}$), $e°(T_{hr})$ is the saturation vapor pressure at air temperature $T_{hr}$ (kPa), $e_a$ is the average hourly actual vapor pressure (kPa), $u_2$ is the average hourly wind speed (m s$^{-1}$), $C_n$ is 37, and $C_d$ is 0.34.

The variation of the FAO Penman-Monteith equation adopted by ASCE and dubbed the ASCE Standardized Reference Evapotranspiration Equation [28] allows $C_n$ and $C_d$ to vary. $ET_0$ calculated for hourly time steps and for a short reference crop during the daytime has $C_n$ = 37 and $C_d$ = 0.24 (giving $ET_{0S}$; Eq 10) and for a tall reference crop during the daytime $C_n$ = 66 and $C_d$ = 0.25 (giving $ET_{0T}$; Eq 11) [28]:

$$ET_{0S} = \frac{0.408\Delta(R_n - G) + \gamma\left(\frac{37}{T_{hr}+273}\right)u_2\left(e^o(T_{hr}) - e_a\right)}{\Delta + \gamma(1 + 0.24 u_2)} \qquad (10)$$

$$ET_{0T} = \frac{0.408\Delta(R_n - G) + \gamma\left(\frac{66}{T_{hr}+273}\right)u_2\left(e^o(T_{hr}) - e_a\right)}{\Delta + \gamma(1 + 0.25 u_2)} \qquad (11)$$

Eqs 9–11 have been used in multiple stormwater studies to model ET in GSI [2, 10, 29]. Additionally, using the PM reference ET equation allows refinement of the expected ET with the application of a crop coefficient.

Crop coefficients for Switchgrass (which are unavailable in the literature) were calculated by methods described in the FAO 56 document as the ratio of mass balance measured ET (Eq 1) to $ET_{0T}$ (Eq 11) [26]:

$$ET_c = K_c ET_{0T} \qquad (12)$$

Values of $K_c$ were derived and compared to those in literature for plants similar to Switchgrass to validate meteorological data and test whether or not the systems studied herein could be described by a common reference ET equation.

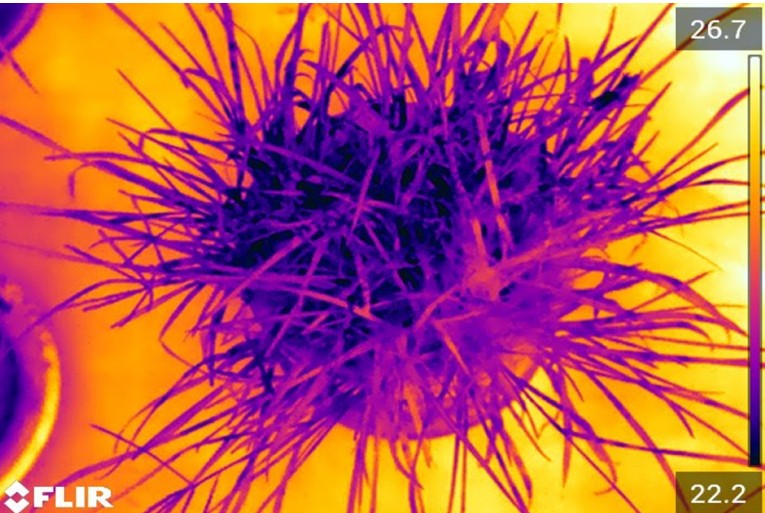

**Fig 2. Example of thermal image from the summer of 2019.** The hottest cell in the image is 26.7˚C and the coolest is 22.2˚C.

**Computational framework for data processing.** Time series weather station and thermal camera data for each collection data in the study period was used to calculate *E* from Eq 8, which yields the mass of water lost through ET. The $T_{canopy}$ (Eqs 5 and 7) was extracted from each matrix of temperature data embedded in the series of thermal images taken at ten second intervals during each data collection day (e.g., **Fig 2**). Image processing to obtain an estimation of leaf area and $T_{canopy}$ was executed via MATLAB R2018a [30]. MATLAB was chosen for its built in image processing packages.

Within the MATLAB scripts for image processing, the first function counted the number of green pixels in the colored image (**Fig 3A**) and cropped that image based on their position (**Fig 3B**). Green pixels were assumed to be plant leaves. Second, the script allowed for spatial calibration to determine the actual size of objects within the colored image. These two tasks together provided the actual green area within an image and was needed to convert ET per area to a mass or volume of water in Eq 8. The second function obtained $T_{canopy}$ to use in Eqs 5 and 7. A representative thermal image is selected (from the ~1,800 thermal images for the data collection period of interest) (**Fig 3C**) and a region of interest (ROI) is drawn over the thermal image, with care that the ROI only contains canopy. That ROI is then used to extract an average $T_{canopy}$ reading from all other thermal images for that data collection event (**Fig 3D**). The third function used the time series meteorological data to carry out calculations for the reference ET equations (Eqs 9–12).

## Results and discussion

The goal of this research was to determine the feasibility of using an energy balance to resolve ET in GSI. For each data collection day, ET was calculated via mass balance (MB) using Eq 1 and Eqs 9, 10, and 11 were used to estimate the reference ET. The average height of the Switch-grass plants in T1, T2, and T3 from August 29, 2019 to October 4, 2019 (0.60 m) was closest to the height of the tall reference crop used to formulate the Eq 11 (alfalfa at 0.50 m). Eq 11 was used to determine crop coefficients (Eq 12) that were compared to literature for other tall grasses [31]. As shown in **Table 1**, ET can be estimated for T1, T2, and T3 using Eq 11 and reasonable crop coefficients [31]. MB measured ET was greater than the reference ET given by

GreenArea.m

AverageTemp.m

**Fig 3. Image processing.** (A) Zoomed in colored image. (B) colored image after thresholding out all non "green" cells and cropping. (C) full thermal image. (D) thermal image cropped by ROI.

Eqs 9–11 for all but two data collection days on bucket T1: September 18, 2019 and September 26, 2019 (Table 1).

The difference between $ET_0$ calculated via the FAO PM (Eq 9) and $ET_{0S}$ (Eq 10) is always less than 0.1 mm, as the two equations were formulated for roughly the same reference crop, similar to an extensive surface of green grass with a uniform height of 0.12 m, that is well-

**Table 1. Comparison between mass balance measured ET and the selected reference ET equations.**

| | | ET | Reference ET (mm) | | | |
|---|---|---|---|---|---|---|
| Column | Date | MB (mm) | FAO PM (Eq 9) | $ET_{0S}$ (Eq 10) | $ET_{0T}$ (Eq 11) | Kc |
| T1 | 8/29/2019 | 4.35 | 2.73 | 2.81 | 2.97 | 1.46 |
| T1 | 9/4/2019 | 3.45 | 2.22 | 2.28 | 2.42 | 1.43 |
| T1 | 9/18/2019 | 1.79 | 1.89 | 1.96 | 2.07 | 0.86 |
| T1 | 9/26/2019 | 1.74 | 1.61 | 1.65 | 1.75 | 0.99 |
| T2 | 8/30/2019 | 4.07 | 2.64 | 2.71 | 2.86 | 1.42 |
| T2 | 9/5/2019 | 2.97 | 1.95 | 2.01 | 2.13 | 1.39 |
| T2 | 9/19/2019 | 2.78 | 2.16 | 2.23 | 2.36 | 1.18 |
| T2 | 10/1/2019 | 2.37 | 1.98 | 2.02 | 2.15 | 1.10 |
| T3 | 9/3/2019 | 5.03 | 2.57 | 2.62 | 2.78 | 1.81 |
| T3 | 9/17/2019 | 4.1 | 2.37 | 2.45 | 2.59 | 1.58 |
| T3 | 9/24/2019 | 2.52 | 1.80 | 1.86 | 1.96 | 1.28 |
| T3 | 10/4/2019 | 2.81 | 1.88 | 1.96 | 2.07 | 1.36 |
| | Average | 3.16 | 2.15 | 2.21 | 2.34 | 1.32 |
| | Standard Deviation | 1.04 | 0.36 | 0.37 | 0.39 | 0.26 |

$K_c$ values were calculated using $ET_{0T}$ from Eq 11. By Eq 12, $K_c$ multiplied by the $ET_{0T}$ gives the MB measured ET.

watered, and completely covering the soil [28]. As expected, $ET_{0T}$ is always greater than both $ET_0$ and $ET_{0S}$. The crop coefficients calculated for Switchgrass (**Table 1**) are reasonable considering crop coefficients in the literature for other tall grasses range from 0.68 to 1.93 [31]. Lastly, crop coefficients for all three buckets are greater earlier in the experimental period (early to mid-September) than later (late September to October), which could be indicative of the Switchgrass plants moving away from a mid-season growth pattern and towards senescence like the transition from $K_{mid}$ to $K_{end}$ suggested in Allen et al. [26]. The available reference ET methods yielded reasonable values compared to the MB measured ET, which indicates that the measured data is within typical expectations and that the available reference equations can be reliably used to give a scalable (via crop coefficient) estimate of ET. Further, this indicates the meteorological data collected and used in the reference ET equation can also be used with confidence in an energy balance method.

For the energy balance, the canopy area was calculated by the first MATLAB function (**Table 2**). The inaccuracies of the method used to obtain canopy area are evident by considering the range of calculated values for T3, for example. One might expect precise measurements of canopy area to remain fairly constant from September 3$^{rd}$ to September 24$^{th}$ or maybe exhibit a decreasing trend from September 3$^{rd}$ to October 4$^{th}$ as the plant went into senescence. However, these results are not observed in the data.

With the canopy area calculated, ET was estimated assuming the view factors for long wave radiation ($F_a = 1$ and $F_g = 0$; [23]). As can be seen in **Table 3** and **Fig 4**, the EB tended to overestimate the ET as calculated by the mass balance, and on two days EB estimated ET was overestimated by more than 2 mm (63% of average MB measured ET). Over the course of 12 data collection events, the average ET estimated by the EB was within 0.96 mm (or 30%) of the MB measured ET without any calibration being done to parameters in the EB. Comparing that 30% difference to the difference between the MB measured ET and $ET_{0T}$ calculated by Eq 11 and given in **Table 1** (3.16–2.34 = 0.82 mm, or 26%) demonstrates the reasonable accuracy of the method developed in this study. In general, especially considering the simple approach taken for resolving the view factors, the energy balance produces results that could be useful to provide a rough estimate of ET rates.

Estimating the radiative exchange between the interior parts of the canopy or the soil and the canopy, and thus partitioning the view factors between $F_a$ and $F_g$, is possible but is beyond the scope of this study. However, a brief exploration into other values for view factors was done to see if the EB estimated ET could provide a closer match to the MB measured ET if

**Table 2.  Canopy Area (A) obtained by GreenArea.m for each data collection day.**

| Column | Date | A (m$^2$) |
|--------|------|-----------|
| T1 | 8/29/2019 | 0.058 |
| T1 | 9/4/2019 | 0.083 |
| T1 | 9/18/2019 | 0.052 |
| T1 | 9/26/2019 | 0.055 |
| T2 | 8/30/2019 | 0.074 |
| T2 | 9/5/2019 | 0.106 |
| T2 | 9/19/2019 | 0.057 |
| T2 | 10/1/2019 | 0.061 |
| T3 | 9/3/2019 | 0.124 |
| T3 | 9/17/2019 | 0.156 |
| T3 | 9/24/2019 | 0.096 |
| T3 | 10/4/2019 | 0.152 |

**Table 3. Energy balance results.**

| Column | Date | Fa | Fg | Mass Balance (mm) | Energy Balance (mm) | Percent Difference (%) |
|---|---|---|---|---|---|---|
| T1 | 8/29/2019 | 1.00 | 0.00 | 4.35 | 3.23 | -29.34 |
| T1 | 9/4/2019 | 1.00 | 0.00 | 3.45 | 3.58 | 3.67 |
| T1 | 9/18/2019 | 1.00 | 0.00 | 1.79 | 2.15 | 18.39 |
| T1 | 9/26/2019 | 1.00 | 0.00 | 1.74 | 1.87 | 7.52 |
| T2 | 8/30/2019 | 1.00 | 0.00 | 4.07 | 3.78 | -7.33 |
| T2 | 9/5/2019 | 1.00 | 0.00 | 2.97 | 4.45 | 39.84 |
| T2 | 9/19/2019 | 1.00 | 0.00 | 2.78 | 2.63 | -5.47 |
| T2 | 10/1/2019 | 1.00 | 0.00 | 2.37 | 2.67 | 11.92 |
| T3 | 9/3/2019 | 1.00 | 0.00 | 5.03 | 6.58 | 26.74 |
| T3 | 9/17/2019 | 1.00 | 0.00 | 4.10 | 7.73 | 61.49 |
| T3 | 9/24/2019 | 1.00 | 0.00 | 2.52 | 3.76 | 39.51 |
| T3 | 10/4/2019 | 1.00 | 0.00 | 2.81 | 6.97 | 85.22 |
| | | | Average | 3.16 | 4.12 | 21.01 |
| | | | Standard Deviation | 1.04 | 1.95 | |

refinement of the view factors was investigated. Refined view factors would give the energy balance the ability to reconcile the fact that the canopy only receives a fraction of the longwave radiation it would receive if it was a flat absorbing surface directly facing the sources of long wave radiation (the atmosphere and the ground). For example, imagine the difference in radiation received by a blade of grass with its broad side facing the sky or ground vs the radiation received by a blade of grass with an edge facing the sky or ground. The initial values used in this study for $F_a$ and $F_g$ to investigate the partitioning were modified from a study done on single soybean leaves oriented parallel to the ground [15]. These values were used to investigate possible refinement of the $F_a$ and $F_g$ since they were the most readily available in literature. View factor values were adjusted from the values for a single soybean leaf ($F_a = 0.80$, $F_g = 0.20$) until the ET produced by the EB closely matched the ET measured via MB (**Table 4**).

Although combinations of view factors that do not add to one are not physically possible this analysis indicates that the energy balance method can produce more reasonable results with values for the view factors in place that take into account the radiative exchange between the canopy and itself, the soil, and other parts of its surroundings besides the atmosphere.

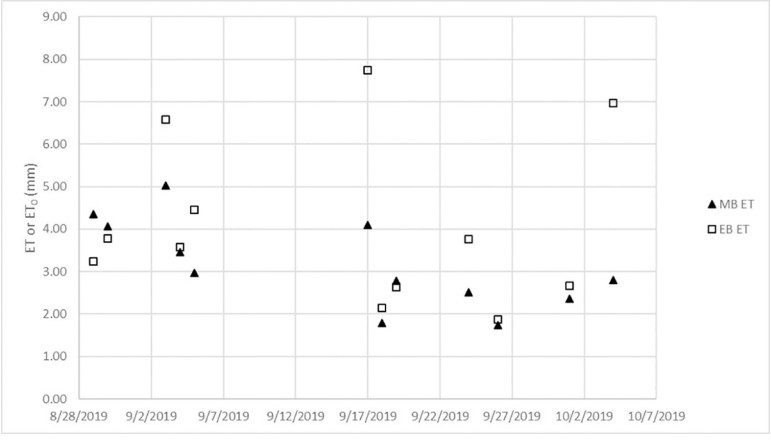

**Fig 4. Energy balance results.**

**Table 4. View factors for each data collection day that produced EB results matching MB ET.**

| Column | Date | View Factors | | ET | |
|--------|------|------|------|------|------|
| | | Fa | Fg | MB (mm) | EB (mm) |
| T1 | 8/29/2019 | 0.71 | 0.55 | 4.35 | 4.34 |
| T1 | 9/4/2019 | 0.72 | 0.21 | 3.45 | 3.45 |
| T1 | 9/18/2019 | 0.62 | 0.17 | 1.79 | 1.78 |
| T1 | 9/26/2019 | 0.65 | 0.25 | 1.74 | 1.74 |
| T2 | 8/30/2019 | 0.71 | 0.16 | 4.07 | 4.07 |
| T2 | 9/5/2019 | 0.56 | 0.10 | 2.97 | 2.97 |
| T2 | 9/19/2019 | 0.71 | 0.26 | 2.78 | 2.78 |
| T2 | 10/1/2019 | 0.60 | 0.25 | 2.37 | 2.37 |
| T3 | 9/3/2019 | 0.57 | 0.10 | 5.03 | 5.03 |
| T3 | 9/17/2019 | 0.26 | 0.11 | 4.10 | 4.09 |
| T3 | 9/24/2019 | 0.57 | 0.11 | 2.52 | 2.52 |
| T3 | 10/4/2019 | 0.12 | 0.10 | 2.81 | 2.80 |
| | Average | 0.57 | 0.20 | 3.16 | 3.16 |

Therefore, if a method was developed for estimating these view factors for GSI, then implementation of a similar energy balance could be possible to obtain more accurate field estimates of ET than given by assuming $F_a = 1$ and $F_g = 0$ (as done in this study).

## Summary and recommendations

The results of this study as a whole demonstrate the possible utility of estimating ET from GSI using an energy balance. They also suggest that given meteorological data collected in very close proximity to GSI and at small (10 second) time steps, common reference ET equations can be used to estimate ET. The difference in input data for the two methods is significant. Reference ET equations do not require images of any kind to be taken of the GSI. However, it is unknown how these reference ET equations would hold up for different GSI systems and more research is needed on crop coefficients for typical GSI vegetation. The energy balance, as formulated herein, should be transferable to any system as it utilizes direct measurements of the energy entering and exiting the system and useful results are obtainable even without any calibration or scaling.

Based on the results of this study, it is evident that through some combination or expansion (or both) of methods presented here and in other literature, ET can and should be accounted for when considering the stormwater volume reduction capabilities of GSI. It is feasible that commonly collected climate data can be used in conjunction with thermal measurements of canopy temperature in GSI to approximate ET. Similar to watershed scale energy balances via satellite thermal imaging, the method derived here could be built upon and used to estimate the water balance and performance of vegetation in GSI. Energy balance methods, like the one described here, may be the only practical way to measure ET in GSI, and therefore fully resolve the water balance in these systems. This research suggests that effort put toward improving the energy balance methods developed here may result in a practical method for measuring the ET performance of full scale GSI.

### Beyond feasibility: Improvements and recommendations

There are several improvements that could be made to this energy balance formulation and the use of the reference ET equations, typically at the cost of data collection and computational time.

One of the large downfalls of the energy balance method described here is the approximation of the canopy area that is actively transpiring water. The issues with the approximation are mainly:

1. The camera is only viewing some fraction of the actual leaf area.

2. The plant canopy is not all contained in the same plane as the object used to spatially calibrate the image. Therefore, the area of parts of the canopy that are closer to the camera than the object used to spatially calibrate the picture is over predicted while the area of the parts of the canopy further from the camera is under predicted. This may be one of the reasons why the energy balance presented over predicts ET. Spatial calibration was performed with an object placed at the base of the plant for each data collection day. Green area was over predicted then since the majority of the plant was closer to the camera than the object used for spatial calibration.

In addition, the equation for the boundary layer conductance to heat $g_{b,h}$ used here is a simplified way to describe the phenomena resisting the transfer of heat through the leaf boundary layer to the bulk atmosphere [15]. Addressing that simplification could help refine the methods used in this work.

The inaccuracies due to placement of the object used for spatial calibration could be addressed by placing the object halfway between the top and bottom of the canopy. Overprediction and underprediction of green area might better negate each other then. Additionally, there could be methods formulated to take that surface area and estimate the three-dimensional canopy surface area from it. This may also allow for actual calculation of $F_a$ and $F_g$ [23]. In the methods used here, the estimated green area is basically projected onto a flat, horizontal surface in the plane of whatever object was used to spatially calibrate the colored image. Accurate measurement of canopy surface area could also improve the reference ET estimates since Eqs 9–11 assume Leaf Area Index (LAI) of a reference surface, when in reality the LAI of the systems in T1, T2, and T3 were likely different (Allen et al. 1998).

The method used for extracting only pixels in the thermal images containing leaf temperature also has room for improvement:

1. It is difficult to tell when drawing an ROI over a thermal image what is canopy and what is not, especially if the soil beneath the canopy is near the same temperature as the leaves.

2. It is not certain that the average temperature of the cells in the ROI are a good representation of the temperature of the entire canopy. Transpiration is not spread evenly through a canopy and there is not one layer to a canopy. Leaves that are receiving the most energy from the environment and are most exposed to wind will carry out the majority of the ET.

Still, this method for extracting relevant temperature data from the images should give a decent approximation of canopy temperature because the ROI drawn for each data collection day captured presumably the most photosynthetically active parts of the leaf canopy without including parts of the image that were not leaf canopy.

Other research areas would be possible if concurrent colored images were taken by the thermal camera. Then, the thermal image could be cropped with a mask created by the concurrent colored image [15]. This would eliminate error from hand drawing the ROI. Another improvement could be made by ensuring reference surfaces are visible to the thermal camera for each time step. This is a concept used when implementing energy balances using thermal imaging from satellites to bound the estimated ET by 0, which corresponds to the hottest/driest cell in the image, and potential ET, which corresponds to the coldest/wettest cell in the image [11]. Implementing this concept would require some area being viewed by the camera

to be non-transpiring and some area to be saturated for each time step. If the reference cell method was implemented in this study, large errors in the EB for days like 9/17 and 10/4 in **Fig 4** could be eliminated. As discussed previously, any method for estimating view factors from easily obtained physical measurements of the GSI system would be an improvement. Concepts for formulating few factors can be found in An Introduction to Environmental Biophysics [23]. The methods are beyond the scope of this study.

Further, it is possible that the methods presented here could be adapted to give the most practical field measurement of ET from GSI. For example, as seen in **Fig 1**, the instrumentation used to collect the necessary data for computing an energy balance could be made to be relatively mobile. If a similar set up was constructed temporarily at the major GSI in a municipal network for instance, a refined version of the energy balance presented here, maybe even along with a reference ET equation, could be implemented to give an estimate of GSI performance.

The goals of this exploratory analysis were met and seed future potential research. This simplified version of the energy balance required less data collection than would have been needed to obtain more accurate estimates of leaf area or view factors. It also required less data processing and the associated computational time than would have been required to perfectly overlay concurrent colored and thermal images, for instance. What this study accomplishes is demonstrating that there may be justification to spend substantially more time refining this energy balance and the methods of obtaining ET from GSI with it.

## Author Contributions

**Conceptualization:** Devin Thomas Krasowski, Bridget Wadzuk, Brian Jacko.

**Data curation:** Devin Thomas Krasowski, Brian Jacko.

**Formal analysis:** Devin Thomas Krasowski, Brian Jacko.

**Funding acquisition:** Bridget Wadzuk.

**Investigation:** Devin Thomas Krasowski, Brian Jacko.

**Methodology:** Devin Thomas Krasowski, Bridget Wadzuk.

**Project administration:** Devin Thomas Krasowski, Bridget Wadzuk.

**Resources:** Devin Thomas Krasowski, Bridget Wadzuk.

**Software:** Devin Thomas Krasowski, Brian Jacko.

**Supervision:** Bridget Wadzuk.

**Validation:** Devin Thomas Krasowski, Bridget Wadzuk.

**Visualization:** Devin Thomas Krasowski.

**Writing – original draft:** Devin Thomas Krasowski.

**Writing – review & editing:** Bridget Wadzuk, Brian Jacko.

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
