## [Decision Letter · Decision Letter 0]

24 Sep 2020

PONE-D-20-27580

Feasibility of using an energy balance to measure evapotranspiration in green stormwater infrastructure

PLOS ONE

Dear Dr. Krasowski,

Thank you for submitting your manuscript to PLOS ONE. After careful consideration, we feel that it has merit but does not fully meet PLOS ONE’s publication criteria as it currently stands. Therefore, we invite you to submit a revised version of the manuscript that addresses the points raised during the review process.

I suggest to consider the comments about the robustness of the experimental methodology, you have to provide a proper discussion on this subject in order to convince the reviewers about the validity of the experiment. Your paper got a "rejection" and a "major revision". You are almost in the middle, so you have to provide adequate explanations about the experimental methodology. Next review does guarantee the acceptance of your paper. 

We look forward to receiving your revised manuscript.

Kind regards,

Vassilis G. Aschonitis

Academic Editor

PLOS ONE

Journal Requirements:

'This work was funded by a grant applied for by and awarded to Dr. Bridget Wadzuk (BW). This internal grant from the Villanova University Department of Civil and Environmental Engineering (https://www1.villanova.edu/university/engineering/academic-programs/departments/civil-environmental.html) was awarded to fund tuition and provide a stipend for one Master's student (DTK) to carry out the research and to provide compensation to undergraduate research assistants (BJ). '

We note that one or more of the authors are employed by a commercial company: SPF Water Engineering, LLC

Additional Editor Comments:

I suggest to consider the comments about the robustness of the experimental methodology, you have to provide a proper discussion on

this subject in order to convince the reviewers about the validity of the experiment. Your paper got a "rejection" and a "major revision".

You are almost in the middle, so you have to provide adequate explanations about the experimental methodology. Next review does guarantee

the acceptance of your paper.

Reviewers' comments:

Reviewer's Responses to Questions

**Comments to the Author**

1. Is the manuscript technically sound, and do the data support the conclusions?

Reviewer #1: Partly

Reviewer #2: Partly

2. Has the statistical analysis been performed appropriately and rigorously? 

Reviewer #1: Yes

Reviewer #2: No

3. Have the authors made all data underlying the findings in their manuscript fully available?

Reviewer #1: Yes

Reviewer #2: Yes

4. Is the manuscript presented in an intelligible fashion and written in standard English?

Reviewer #1: Yes

Reviewer #2: Yes

5. Review Comments to the Author

Reviewer #1: The manuscript is enough well-written, but the language can be improved. Although the bibliography has a range of time, there are points in the manuscript that are not well justified such as lines 71-74 that are missing of references. Furthermore, I have an objection with the duration of the experiment and the number. Maybe another one for more accuracy is needed.

It is mentioned in the text that there are three equations that are being used for the ET. Are there? It is not clear enough. The authors chose to use MATLAB. What is the reason for? Why not something else?

Do the authors take into consideration that the camera is in a stable position and that may cause a shading and gives other results than the expected ones?

In line 363 it is not clear how the soybean and the switchgrass are having the same view factors. Generally it is not well written and clear how the authors have chosen the view factors.

Moreover, are the conditions the same in order to have each lysimeter in a test position?

In line 377, the statement needs a better justification.

In line 385, it is said that there is no need of calibration. I have an objection and it is expendient to be done a better search in the scientific bibliography.

In line 428-429, the sentence causes a lot of questions about the experiment and the results.

In conclusion, the manuscript needs an overall overview by the authors in order to explain in a better way many aspects in the text, especially the whole experiment, the lysimeters , the average height of the plant.

Reviewer #2: Feasibility of using an energy balance to measure evapotranspiration in green stormwater infrastructure

This manuscript examines the feasibility of using energy balance models including thermal images for the calculation of ET in GSIs. The precise calculation of ET in GSIs is quite important in treatment measure determination for the decrease of floods possibility in urban areas. The general investigation has clear construction but I do have some severe concerns mainly in adequacy of data set and experimental procedure.

i) Line 76: “arewidely” space is missing.

ii) Lines 81, 395-396: You could make “Green stormwater infrastructure” to be “GSI” as you do in other cases. I suggest to define the abbreviation GSI in the first mention (line 33) and then you can call it as GSI.

iii) Lines 84-85: “Energy balance methods have not been investigated to calculate ET for GSI.” You should probably re-examine or rephrase that, from a little search I did, I found a paper in which the Eddy-covariance method is investigated in green infrastructure. Here is the link of the paper:

https://www.sciencedirect.com/science/article/pii/S0048969716323567?via%3Dihub

iv) Figure 1: I see that the heights of the plants in lysimeters are quite different. It is on purpose? If it was on purpose, what is the point? I think this requires more explanation. Having in mind the consideration of ASCE for tall reference crop, do you think this difference of the height between the three plants (if it’s real) influences the results? In lines 286-288 is referred “The average height of the Switchgrass plants in T1, T2, and T3 from August 29, 2019 to October 4, 2019 (0.60m)”, the average of three plants for the entire period (August 29, 2019 to October 4)?

v) Table 1: In the 3 columns of “Reference ET” units are missing.

vi) Lines 330-331: “Over the course of 12 data collection events, the average ET estimated by the EB was within 1 mm (or 32%) of the MB measured ET and there was no calibration done to any parameters in the EB.” I do not exactly understand what is the meaning here.

vii) Concerning the data set, do you think is large enough for the calibration of Fa and Fg coefficients requirements? Additionally, what about the validation of the energy balance model?

viii) Line 394: “An energy balance like the one described here may be the only practical way to measure ET in green stormwater infrastructure, and thus, the only way to fully resolve the water balance in these systems.” I am not sure about the meaning and the correctness of this.

6. PLOS authors have the option to publish the peer review history of their article (what does this mean?). If published, this will include your full peer review and any attached files.

Reviewer #1: No

Reviewer #2: No

---

## [Author Response · Author response to Decision Letter 0]

8 Nov 2020

All responses should be in the Response to Reviewers file. Thanks again.

---

## [Decision Letter · Decision Letter 1]

27 Nov 2020

PONE-D-20-27580R1

Feasibility of using an energy balance to measure evapotranspiration in green stormwater infrastructure

PLOS ONE

Dear Dr. Krasowski,

Thank you for submitting your manuscript to PLOS ONE. After careful consideration, we feel that it has merit but does not fully meet PLOS ONE’s publication criteria as it currently stands. Therefore, we invite you to submit a revised version of the manuscript that addresses the points raised during the review process.

We look forward to receiving your revised manuscript.

Kind regards,

Vassilis G. Aschonitis

Academic Editor

PLOS ONE

Reviewers' comments:

Reviewer's Responses to Questions

**Comments to the Author**

1. If the authors have adequately addressed your comments raised in a previous round of review and you feel that this manuscript is now acceptable for publication, you may indicate that here to bypass the “Comments to the Author” section, enter your conflict of interest statement in the “Confidential to Editor” section, and submit your "Accept" recommendation.

Reviewer #2: All comments have been addressed

Reviewer #3: All comments have been addressed

Reviewer #4: All comments have been addressed

2. Is the manuscript technically sound, and do the data support the conclusions?

Reviewer #2: Yes

Reviewer #3: Yes

Reviewer #4: Yes

3. Has the statistical analysis been performed appropriately and rigorously? 

Reviewer #2: Yes

Reviewer #3: Yes

Reviewer #4: N/A

4. Have the authors made all data underlying the findings in their manuscript fully available?

Reviewer #2: Yes

Reviewer #3: Yes

Reviewer #4: Yes

5. Is the manuscript presented in an intelligible fashion and written in standard English?

Reviewer #2: Yes

Reviewer #3: Yes

Reviewer #4: Yes

6. Review Comments to the Author

Reviewer #2: I read the original and the revision submission and I found that many corrections have been done and the responses to the reviewer were satisfactory. I have only one objection.

Lines 441 – 442: “may be the only practical 441 way to measure ET GSI, and thus, the only way to fully resolve the water balance in these systems.” I think you could use some milder words (“only way”).

Reviewer #3: The manuscript developed a new procedure for estimating ET in bioretention system. It is well structured, and the objectives are clear. Therefore, I suggested that it could be accepted for publication.

Reviewer #4: I reviewed the initial version and the version of the revision of this manuscript. I think there are significant improvements in this manuscript after doing the revisions according to reviewers suggestions. However, in my opinion there is still some questions. I have listed my suggestions below.

1. In Line 129, although the authors have pointed out that “Three lysimeters contained Switchgrass (labeled T1 through T3)”, I still do not understand what T1, T2, T3 mean. I think the authors should describe them clearly.

2. In Line 347-348, “the average ET estimated by the EB was within 1 mm (or 32%) of the MB measured ET without any calibration being done to parameters in the EB. This small difference.…..”. In fact, I think “32%” is not a small difference.

3. In Line 394: “An energy balance like the one described here may be the only practical way to measure ET in green stormwater infrastructure, and thus, the only way to fully resolve the water balance in these systems.” I am not sure about the meaning and the correctness of this.

4. In Line 411-413: “Energy balance methods, like the one described here, may be the only practical way to measure ET in GSI, and thus, the only way to fully resolve the water balance in these systems.” As suggested by the former reviewer, I am also not sure about the meaning and the correctness of this. I think the method should not be the only way in GSI. Please consider and correct it.

7. PLOS authors have the option to publish the peer review history of their article (what does this mean?). If published, this will include your full peer review and any attached files.

Reviewer #2: No

Reviewer #3: No

Reviewer #4: No

---

## [Author Response · Author response to Decision Letter 1]

10 Dec 2020

Please see attached files addressing review comments and questions.

---

## [Decision Letter · Decision Letter 2]

13 Jan 2021

PONE-D-20-27580R2

Feasibility of using an energy balance to measure evapotranspiration in green stormwater infrastructure

PLOS ONE

Dear Dr. Krasowski,

Thank you for submitting your manuscript to PLOS ONE. After careful consideration, we feel that it has merit but does not fully meet PLOS ONE’s publication criteria as it currently stands. Therefore, we invite you to submit a revised version of the manuscript that addresses the points raised during the review process.

We look forward to receiving your revised manuscript.

Kind regards,

Vassilis G. Aschonitis

Academic Editor

PLOS ONE

Reviewers' comments:

Reviewer's Responses to Questions

**Comments to the Author**

1. If the authors have adequately addressed your comments raised in a previous round of review and you feel that this manuscript is now acceptable for publication, you may indicate that here to bypass the “Comments to the Author” section, enter your conflict of interest statement in the “Confidential to Editor” section, and submit your "Accept" recommendation.

Reviewer #2: All comments have been addressed

Reviewer #4: (No Response)

2. Is the manuscript technically sound, and do the data support the conclusions?

Reviewer #2: Yes

Reviewer #4: Yes

3. Has the statistical analysis been performed appropriately and rigorously? 

Reviewer #2: Yes

Reviewer #4: Yes

4. Have the authors made all data underlying the findings in their manuscript fully available?

Reviewer #2: Yes

Reviewer #4: Yes

5. Is the manuscript presented in an intelligible fashion and written in standard English?

Reviewer #2: Yes

Reviewer #4: Yes

6. Review Comments to the Author

Reviewer #2: (No Response)

Reviewer #4: Although the authors tried to improve the manuscript and explain the Reviewer’s comment. However, I think the authors have not made some substantial changes according to the reviewers' comments. For example:

1.I still do not understand what T1, T2, T3 mean. Is there a difference among these three treatments? If yes, what is the difference? I think the authors still did not describe them clearly.

2.In line 347-351, whether the difference is small should be evaluated quantitatively by using statistical methods, rather than qualitative judgments.

7. PLOS authors have the option to publish the peer review history of their article (what does this mean?). If published, this will include your full peer review and any attached files.

Reviewer #2: No

Reviewer #4: No

---

## [Author Response · Author response to Decision Letter 2]

15 Jan 2021

Please see attached Response to Reviewers.

---

## [Editor Report · Decision Letter 3]

18 Jan 2021

Feasibility of using an energy balance to measure evapotranspiration in green stormwater infrastructure

PONE-D-20-27580R3

Dear Dr. Krasowski,

We’re pleased to inform you that your manuscript has been judged scientifically suitable for publication and will be formally accepted for publication once it meets all outstanding technical requirements.

Kind regards,

Vassilis G. Aschonitis

Academic Editor

PLOS ONE
---

## [Editor Report · Acceptance letter]

21 Jan 2021

PONE-D-20-27580R3 

Feasibility of using an energy balance to measure evapotranspiration in green stormwater infrastructure 

Dear Dr. Krasowski:

I'm pleased to inform you that your manuscript has been deemed suitable for publication in PLOS ONE. Congratulations! Your manuscript is now with our production department. 

Kind regards, 

on behalf of

Dr. Vassilis G. Aschonitis 

Academic Editor

PLOS ONE